# Effective interventions for improving routine childhood immunisation in low and middle-income countries: a systematic review of systematic reviews

Monica Jain ,[1] Maren Duvendack,[2] Shannon Shisler,[3] Shradha S Parsekar ,[1] Maria Daniela Anda Leon[3]

[1]International Initiative for Impact Evaluation, New Delhi, Delhi, India
[2]University of East Anglia, Norwich, UK
[3]International Initiative for Impact Evaluation, Washington, DC, USA

**Correspondence to**
Dr Monica Jain;
mjain@3ieimpact.org

## ABSTRACT

**Objective** An umbrella review providing a comprehensive synthesis of the interventions that are effective in providing routine immunisation outcomes for children in low and middle-income countries (L&MICs).

**Design** A systematic review of systematic reviews, or an umbrella review.

**Data sources** We comprehensively searched 11 academic databases and 23 grey literature sources. The search was adopted from an evidence gap map on routine child immunisation sector in L&MICs, which was done on 5 May 2020. We updated the search in October 2021.

**Eligibility criteria** We included systematic reviews assessing the effectiveness of any intervention on routine childhood immunisation outcomes in L&MICs.

**Data extraction and synthesis** Search results were screened by two reviewers independently applying predefined inclusion and exclusion criteria. Data were extracted by two researchers independently. The Specialist Unit for Review Evidence checklist was used to assess review quality. A mixed-methods synthesis was employed focusing on meta-analytical and narrative elements to accommodate both the quantitative and qualitative information available from the included reviews.

**Results** 62 systematic reviews are included in this umbrella review. We find caregiver-oriented interventions have large positive and statistically significant effects, especially those focusing on short-term sensitisation and education campaigns as well as written messages to caregivers. For health system-oriented interventions the evidence base is thin and derived from narrative synthesis suggesting positive effects for home visits, mixed effects for pay-for-performance schemes and inconclusive effects for contracting out services to non-governmental providers. For all other interventions under this category, the evidence is either limited or not available. For community-oriented interventions, a recent high-quality mixed-methods review suggests positive but small effects. Overall, the evidence base is highly heterogenous in terms of scope, intervention types and outcomes.

**Conclusion** Interventions oriented towards caregivers and communities are effective in improving routine child immunisation outcomes. The evidence base on health system-oriented interventions is scant not allowing us to reach firm conclusions, except for home visits. Large evidence gaps exist and need to be addressed. For

## STRENGTHS AND LIMITATIONS OF THIS STUDY

⇒ This review synthesises the evidence base on routine child immunisation interventions in low and middle-income countries at a higher level of abstraction (ie, at the systematic review of systematic reviews level), thus filling a gap in the literature.

⇒ Our review uses an intervention taxonomy that was developed and refined based on existing literature, extensive discussion, pilot testing and expert feedback allowing an assessment of interventions oriented towards caregivers, health systems, communities as well as those outside the health realm separately.

⇒ Our review synthesises the evidence both quantitatively through meta-analysis and qualitatively, therefore providing insights into where evidence exists and point out critical evidence synthesis gaps.

⇒ The review cautions on the interpretation of our findings because of methodological challenges, including mixed quality of the included systematic reviews and of the primary studies that have included in their synthesis as well as small sample bias.

example, more high-quality evidence is needed for specific caregiver-oriented interventions (eg, monetary incentives) as well as health system-oriented (eg, health workers and data systems) and community-oriented interventions. We also need to better understand complementarity of different intervention types.

## INTRODUCTION

In 2019, close to 20 million children in low and middle-income countries (L&MICs) did not receive the three recommended doses of diphtheria, pertussis and tetanus (DPT) vaccines, often used as an indicator to assess countries' performance on routine immunisation.[1] Ten countries, in particular Nigeria, India, the Democratic Republic of Congo and Pakistan, account for two out of five unvaccinated children globally.[2] Many L&MICs struggle to achieve high immunisation

coverage due to challenges on both the supply and demand side. Examples of supply side constraints include limited availability of health personnel and difficulties in building up their capacity and skills as well as lack of reliable monitoring and surveillance systems to identify and monitor unvaccinated and undervaccinated children.[3] Demand side issues linked to behavioural, social and practical constraints faced by caregivers also play a role, including concerns about vaccine safety, lack of knowledge of when and where to vaccinate their children, fears about vaccination side effects and difficulties in accessing health services.[4] Governments in L&MICs have strengthened national-level immunisation interventions to address supply side constraints and are increasingly focusing on addressing constraints faced by caregivers and communities. To support policymakers and practitioners in making evidence-informed decisions on the interventions that have been effective in addressing the barriers to routine vaccination uptake, it is imperative to provide systematically synthesised evidence.

Several methodologically robust impact evaluations and systematic reviews evaluating the effectiveness of these approaches have been published in the past decade.[5–12] However, there has been no attempt yet to synthesise this extensive evidence base at the meta-level with the exception of Heneghan *et al*[13] and Besnier *et al*.[14] First, Heneghan *et al* qualitatively synthesise child and adult immunisation interventions in both L&MICs and high-income countries (HICs) finding inconclusive results for L&MICs due to the small number of reviews for these countries, inadequate definitions of interventions and suboptimal reporting of interventions in the reviews. Second, Besnier *et al* examine interventions focusing on improving child health using narrative synthesis tools; they find positive effects of interventions improving immunisation communication, education and social mobilisation on vaccine uptake. While the evidence base on this topic is growing rapidly, Engelbert *et al*[15] demonstrate that there are clear evidence gaps preventing us from fully understanding what type of immunisation intervention works best, for whom, how and where. This is the first systematic review of systematic reviews with the primary goal to provide more clarity on the type of interventions that are effective in improving routine immunisation outcomes for children in L&MICs. Establishing what is and is not known about the effectiveness of such interventions is crucially important to inform policymakers and practitioners.

## METHODS
### Search strategy and selection criteria
This is a systematic review of systematic reviews adopting a mixed-methods approach including a meta-analysis and narrative elements. It draws on the intervention-outcome framework (online supplemental appendix 1) developed by Engelbert *et al*[15] which distinguishes interventions by different levels, that is, 1st, 2nd and 3rd tiers focusing on the supply side (health system oriented)

as well as the demand side (caregiver oriented). It also allows for the identification of community-oriented interventions as well as for those which are non-health policy oriented. This framework guides the presentation of our results. In terms of study design, we included systematic reviews with and without meta-analyses. The population of interest included children below the age of 5 living in L&MICs, but we occasionally included other populations, for example, caregivers and health workers, especially when these were relevant to understanding the impact of immunisation programmes on intermediate outcomes such as attitudes about vaccination, and access to immunisation services. We included studies examining the impact of any intervention on at least one outcome in relation to routine child immunisation. These outcomes are coverage rates or timeliness of full immunisation, third dose of DPT or pentavalent, or measles; additional antigen-specific immunisation coverage outcomes; and intermediate outcomes that precede them in the theoretical causal chain (eg, attitudes about vaccination and access to immunisation services). The detailed eligibility criteria can be found in online supplemental appendix 2. There were no inclusion restrictions by publication status or language, but studies were excluded when they did not meet our definition of systematic reviews, when they were not effectiveness or intervention reviews, or when they only focused on HICs (online supplemental appendix 3).

We adopted a comprehensive search strategy that was initially designed for an evidence gap map (EGM)[15] examining the routine child immunisation sector in L&MICs. The search strategy included electronic searches of academic databases and the grey literature, that is, institutional websites (online supplemental appendix 4). The search was completed in October 2021. All search results were screened on title/abstract by a team of trained reviewers. Two reviewers independently screened each abstract. During title and abstract screening, weekly reconciliation meetings were held to discuss and resolve disagreements. Full text screening followed the same approach.

### Data analysis
Data were extracted by two researchers independently with reconciliation by others in the review team. Any disagreements were resolved by discussion. We extracted data on context, type of intervention, type of review, design and methods used, outcome measures, quality assessment, study results and findings (online supplemental appendix 5). We have only extracted information at the systematic review level.

#### Non-independence of reviews (or overlap)
Non-independence of reviews (or overlap) is an issue unique to systematic reviews of systematic reviews. Overlap explores to what extent the primary studies included in the pool of systematic reviews are the same or different. We assessed overlap by compiling a citation matrix which includes all the primary studies (one per row) included

in the individual systematic reviews (one per column). Primary studies were sorted alphabetically, and duplicates removed. The primary studies that were the same across reviews were ticked with a check mark. This allowed us to calculate the corrected covered area (CCA) index that describes the extent of overlap in per cent terms[16 17]:

$$CCA = \frac{N-r}{(r \times c) - r},$$

where N is the total number of primary studies included in the systematic reviews (the sum of ticked boxes in the citation matrix), r is the number of rows (the primary studies) and c is the number of columns (included systematic reviews). Pieper et al[17] provide criteria for the interpretation of CCA, where 0–5% suggests slight overlap, 6–10% moderate overlap, 11–15% high overlap and >15% very high overlap. Hennessy and Johnson[16] recommend further overlap investigations in case of a heterogenous evidence base, hence we compare subclusters of reviews examining similar outcomes as well.

### Assessment of risk of bias in included studies

The quality of systematic reviews is assessed differently compared with assessing the quality of the primary studies included in systematic reviews; this is due to the unique methodological features of systematic reviews. The quality of the included reviews was assessed independently by two researchers using the checklist developed by the Specialist Unit for Review Evidence (SURE) to allow a critical appraisal to ensure that minimum levels of methodological rigour are met. The SURE[18] checklist was slightly adapted (online supplemental appendix 6).

We also extracted information on the quality assessment tools reported in each included review which have been used to assess the quality of the underlying primary evidence base by the reviews. Nevertheless, risk of bias of primary or supplementary primary studies was not assessed.

### Data synthesis

We adopted a mixed-methods synthesis approach to best accommodate both the quantitative and qualitative information available. To synthesise the quantitative data, we implemented a robust variance estimation (RVE) meta-regression approach[19] as this performs well even when study numbers are as low as 10.[20]

When an RVE meta-regression approach was not possible, we fitted a random-effects model to the data. In this case, the amount of heterogeneity (ie, $\tau^2$), was estimated using the DerSimonian-Laird estimator.[21] In addition, we also reported the Q-test for heterogeneity[22] and the $I^2$ statistic.[23] Studentised residuals and Cook's distances were used to examine whether studies may be outliers and/or influential in the context of the model.[24] Studies with a studentised residual larger than the $100 \times (1 - 0.05 / (2 \times k))^{th}$ percentile of a standard normal distribution were considered potential outliers (ie, using a Bonferroni correction with two-sided $\alpha = 0.05$ for k studies included in the meta-analysis). Studies with a Cook's distance larger than the median plus six times the IQR of the Cook's distances were considered to be influential. The rank correlation test[25] and the regression test,[26] using the SE of the observed outcomes as predictor, were used to check for funnel plot asymmetry but only when we had more than 10 studies.[27]

Risk ratios (RRs) were the most commonly reported effect size, and thus our metric of choice. ORs were converted to RRs using the following formula[28]:

$$RR = OR / (1p + (p \times OR)),$$

where p is the risk in the control group implying that ORs can only be converted to RRs when the risk in the control group is known. As this is not always the case, we established a range of plausible risks for the control group (Grant[28]) drawing on data from Jain et al[29 30] on baseline full immunisation rates to find the mean (0.48) and SD (0.22) of the data to be able to convert ORs to RRs. We did these conversions three times as a sensitivity analysis. First, we used 1 SD below the mean to establish the control group rate (0.26—the least conservative model); second, we used the mean as the control group rate (0.48—the baseline model); and finally, we used 1 SD above the mean (0.70—the conservative model) as the control group rate.

### Patient and public involvement

No patients have been involved in this study as it is a review article, hence no patient consent for publication was required.

### RESULTS

Our search builds on Engelbert et al[15] who had included 58 systematic reviews in their EGM, which we screened against our inclusion criteria. Six of these reviews did not meet our criteria for inclusion and were thus excluded leaving us with 52 studies. We ran additional searches (online supplemental appendix 4); the electronic search led to 1687 records of which 818 were duplicates leaving 869 for screening. We excluded 823 records based on title and abstract screening. The remaining 46 studies were screened by full text, 39 were excluded, leaving seven new studies for inclusion derived from the electronic search. We identified a further seven records searching the grey literature which all required full text screening. Four were subsequently excluded leaving three new studies for inclusion (figure 1). This two-pronged search strategy led to 10 new systematic reviews to be included. Overall, we included a total of 62 systematic reviews—52 studies from the EGM by Engelbert et al[15] and 10 new studies from the updated search. A summary of the included systematic reviews can be found in online supplemental appendix 7.

### Overlap

We test overlap by compiling a citation matrix listing all primary studies which is used to calculate the CCA. Table 1 suggests limited overlap of 0.5% confirming our initial investigations. This is confirmed by Pieper et al[17]

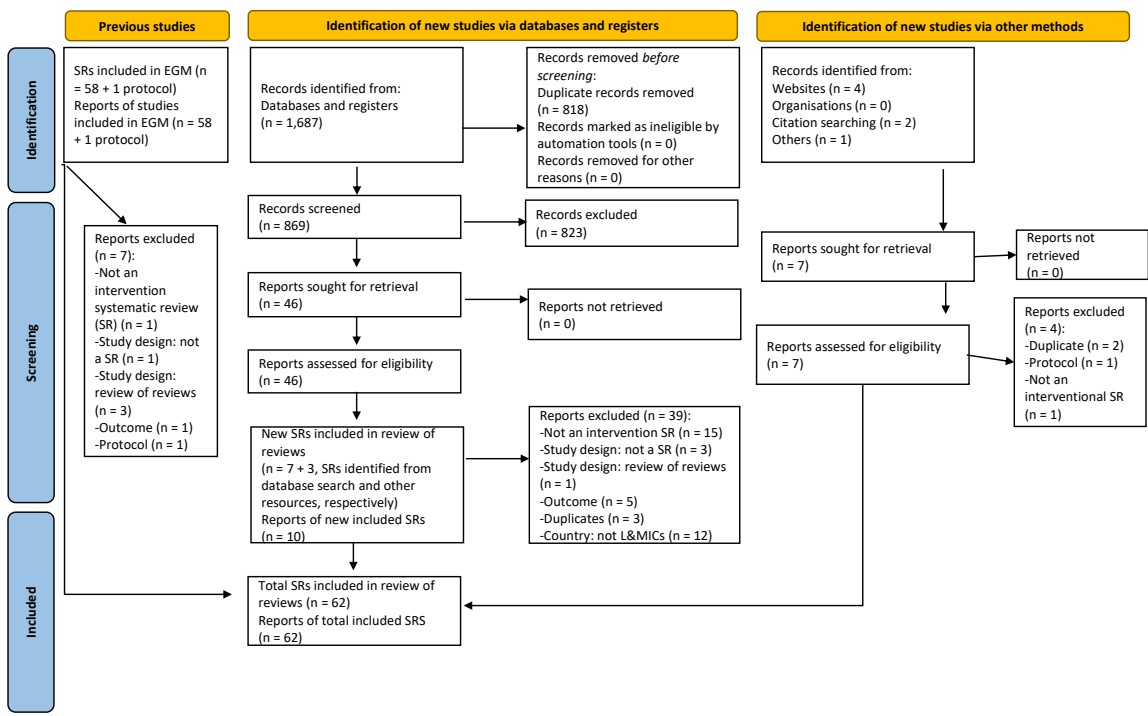

**Figure 1** Preferred Reporting Items for Systematic Reviews and Meta-Analyses (PRISMA) flow diagram. From: Page *et al*[32] and Haddaway *et al*.[33] EGM, evidence gap map; L&MICs, low and middle-income countries.

who provide guidance on the interpretation of CCA values, for example, with values between 0% and 5% indicating slight overlap.

We compare subclusters of reviews examining similar outcomes as well finding limited overlap for most outcomes, for example, for measles, full routine immunisation for children, DPT3 and vaccination coverage, the CCA values range from 0.7% to 2.7%. Vaccination timeliness has the highest CCA value (6.9%) indicating moderate levels of overlap.[17]

### Risk of bias
Using the SURE checklist, we find that of the 62 included reviews, 18 are categorised as high confidence, 6 are of medium confidence and 38 are of low confidence (online supplemental appendix 8).

### Effectiveness of interventions
Overall, the evidence base is highly heterogenous in terms of scope, intervention types and outcomes. Table 2 (and online supplemental appendix 9) summarises the key findings across all high and medium confidence reviews

across all intervention categories. Most of the evidence we uncovered is centred on caregiver-oriented interventions (A) that have positive and statistically significant effects at the first and second tier analysis level (online supplemental appendices 10 and 11 for findings with details on intervention coding and level of analysis in table 2—online supplemental appendix 9). At the third tier level, our analysis confirms the favourable trends emerging from the first and second tier analyses. For short-term sensitisation and education campaigns (AA2—figure 2), we find that the average RR was 1.38 (95% CI 1.33 to 1.44), suggesting that the average outcome differed significantly from zero (z=47.37, p<0.001). As for written or pictorial messages (short message service (SMS), stickers, flyers, etc) to caregivers (AB4—figure 3), we find that the average RR of the baseline model stands at 1.24 (95% CI 1.11 to 1.36, p<0.001). Overall, third tier intervention categories are reporting positive and statistically significant effects, which suggest that treated children are more likely to be vaccinated than untreated children. Another intervention category at the third tier for which there is

| Table 1 | Overlap | | | | |
|---|---|---|---|---|---|
| | Times studies appeared in reviews | Number of rows | Number of reviews | CCA values | |
| | N | r | c | Proportion | Percentage |
| Overall | 1428 | 1079 | 62 | 0.0053 | 0.5 |
| CCA, corrected covered area. | | | | | |

**Table 2** Summary of findings

| Intervention (1st, 2nd, 3rd tiers) | | | SRs (n) | SRs (low confidence SRs in parenthesis) | Overall findings* |
|---|---|---|---|---|---|
| **1st tier** | **2nd tier** | **3rd tier** | | | |
| A. Caregiver-oriented interventions | AA. Information and education | AA1. Sustained sensitisation and education campaigns | 1 | (34) | None (low confidence evidence) |
| | | AA2. Short-term sensitisation and education campaigns | 12 | 6 8 9 11 12 35 (36–41 | Positive and statistically significant |
| | | AA3. Public information campaigns | 3 | (41–43) | None (low confidence evidence) |
| | AB. Incentives and motivation | AB1. Material/monetary incentives for caregivers | 12 | 11 44 45 (36 38 46–52) | Mixed effects |
| | | AB3. Automated voice messages to caregivers | 1 | (53) | None (low confidence evidence) |
| | | AB4. Written or pictorial messages (SMS, stickers, flyers, etc) to caregivers | 13 | 5 10 54–57 (36 38 39 48 53 58 59) | Positive and statistically significant |
| | | AB5. Changes to health system user fees | 1 | (60) | None (low confidence evidence) |
| B. Health system oriented | BA. Education and training | BA1. Formal health worker training and education | 3 | 11 (38 51) | Limited evidence |
| | BB. Planning, implementation, monitoring | BB5. Outreach to vulnerable populations (hard to reach, etc) | 3 | 11 (36 43) | Limited evidence |
| | | BB7. Home visits | 5 | 11 61 (41 62 63) | Positive |
| | BC. Supplementary immunisation activities | BC1. National/subnational immunisation days | 2 | (37 62) | None (low confidence evidence) |
| | BD. Incentives and motivation | BD1. Material/monetary incentives for health workers | 1 | (41) | None (low confidence evidence) |
| | | BD4. Written or pictorial messages to health workers | 1 | (46) | None (low confidence evidence) |
| | | BD5. Pay-for-performance schemes | 5 | 7 64 (46 65 66) | Mixed effects |
| | BF. Health system governance, policies and financing | BF1. Health system strategic planning | 7 | 11 (43 46 48 67–69) | Limited evidence |
| | | BF4. Health system financing | 3 | 70 71 (72) | Inconclusive |
| | BG. Technology and mHealth | BG1. New Health Management Information System (HMIS) / dashboard systems | 3 | (39 73 74) | None (low confidence evidence) |
| C. Other community member oriented | CA. Other community member oriented | CA1. Faith-based outreach/ outreach using local leaders | 1 | (38) | None (low confidence evidence) |
| D. Community level | DA. Communication and dialogue | DA1. Collaborating with whole community | 1 | 75 | Limited evidence |
| | | DA2. Collaborating with selected community groups and networks | 1 | 76 | Limited evidence |
| F. Multicomponent | | | 1† | 29 30 | Positive and statistically significant |

*In the last column, 'overall findings' are summarised from evidence as reported in the systematic reviews, qualitative and quantitative findings, and quantitative findings from review of reviews (RoR). Kindly refer to online supplemental appendix 9 for the expanded version of this table.
†Apart from Jain et al,[29 30] there were 25 systematic reviews that assessed the effects of more than one intervention that are not part of this table. Of these reviews, the majority (18) were low confidence SRs and the intervention components were heterogenous. We provide a summary of Jain et al[29 30] in this table because it is one of the only included SRs that assessed community engagement interventions. Intervention and outcome codes are taken from online supplemental appendix 1.
SMS, short message service; SR, systematic review.

a large evidence base is monetary incentives (AB1) (The evidence on monetary incentive (AB1) interventions is dominated by qualitative narrative methods. There are only two reviews with quantitative evidence, but we could not use them for quantitative synthesis as we were not able to convert effect sizes from standardised mean difference (SMD) to RR) and the evidence is mixed.

For the health system-oriented interventions (B) we had no quantitative evidence, thus we could only rely on narrative synthesis and the results are mixed (online

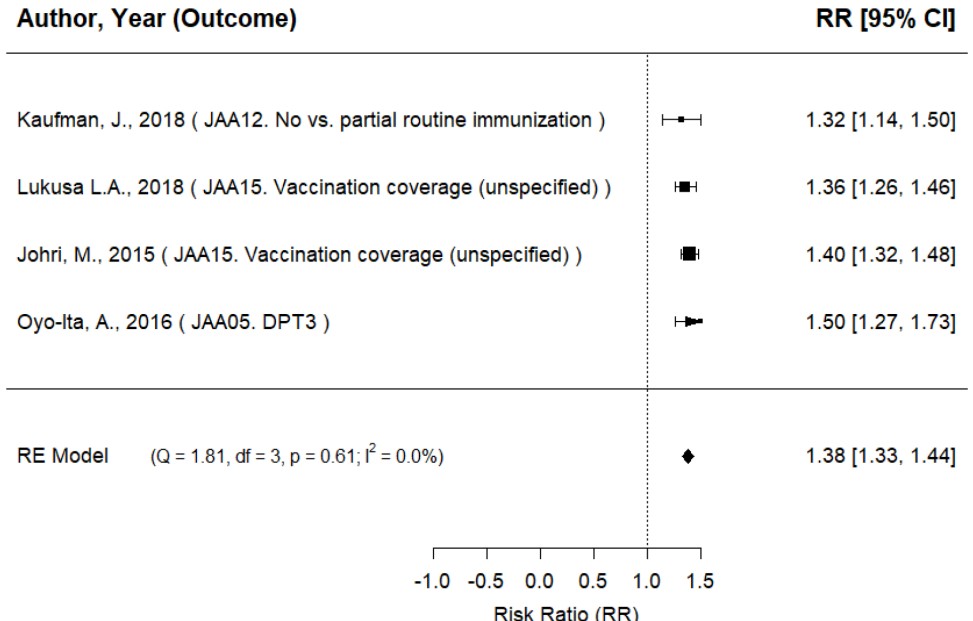

**Figure 2** Random-effects (RE) model of short-term sensitisation and education campaigns (AA2). Intervention and outcome codes are taken from online supplemental appendix 1. DPT, diphtheria, pertussis and tetanus.

supplemental appendix 12). Depending on the exact nature of the intervention, the contextual background and the specific outcome, the findings range from favourable to inconclusive effects. Among these interventions, the only intervention category at the third tier for which we find positive effects is home visits (BB7). We find mixed results for pay-for-performance schemes (BD5) which have a large evidence base both at the primary study and review level. Additionally, we find inconclusive results based on a thin evidence base for health systems financing (BF4) interventions in which health services are contracted out to non-governmental providers and they are compensated for it. For a third tier category like health system strategic planning (BF1), though the evidence base is large when it occurs in combination with other interventions, that is, as multicomponent interventions, the evidence base for single interventions is rather limited and inconclusive.

As for community-oriented interventions (C and D), the evidence base is very limited, focused on narrative synthesis and finding inconclusive results (online supplemental appendix 13). However, a recent review[29 30] on single and multicomponent community engagement interventions (F) uses a nuanced framework to classify them based on the process of engaging communities and finds them to be effective using meta-analytical methods. We found no evidence on interventions related to non-health-related policies and institutions (E).

This review also assessed the interventions aiming to improve zero dose outcomes through DPT1 and BCG vaccine uptake. We find four reviews suggesting inconclusive evidence for interventions improving DPT1 outcomes. However, for BCG vaccine uptake we find pay-for-performance schemes (BD5—one qualitative review)

and community engagement interventions (C and D—one quantitative review) to be effective.

## DISCUSSION

Through this review we have demonstrated that a wealth of systematic reviews on interventions impacting routine child immunisation outcomes exists, but many are very broad in their scope, and hence may not provide a clear answer on all the different types of interventions that may or may not work best for improving immunisation outcomes of children in L&MICs.

For caregiver-oriented interventions (A) the evidence base is of reasonable quality and sufficiently large compared with that on health systems (B) and community-oriented (C and D) interventions. We also find that caregiver-oriented interventions, like those focusing on short-term sensitisation and education campaigns (AA2) as well as written messages to caregivers (AB4), are effective with most findings suggesting positive and statistically significant results. On the other hand, for health system-oriented interventions the evidence base is thin with narrative synthesis approaches dominating and suggesting inconclusive results. For community-oriented interventions the evidence base is limited, but a recent review[29 30] provides high-quality evidence on the effectiveness of community engagement interventions with mostly positive and statistically significant results. Overall, the evidence base provides clearer answers on the effectiveness of caregiver-oriented interventions and to a certain degree of community-oriented interventions than of health system-oriented interventions.

### Strengths and weaknesses
We have contributed to the literature by synthesising the evidence base on routine child immunisation

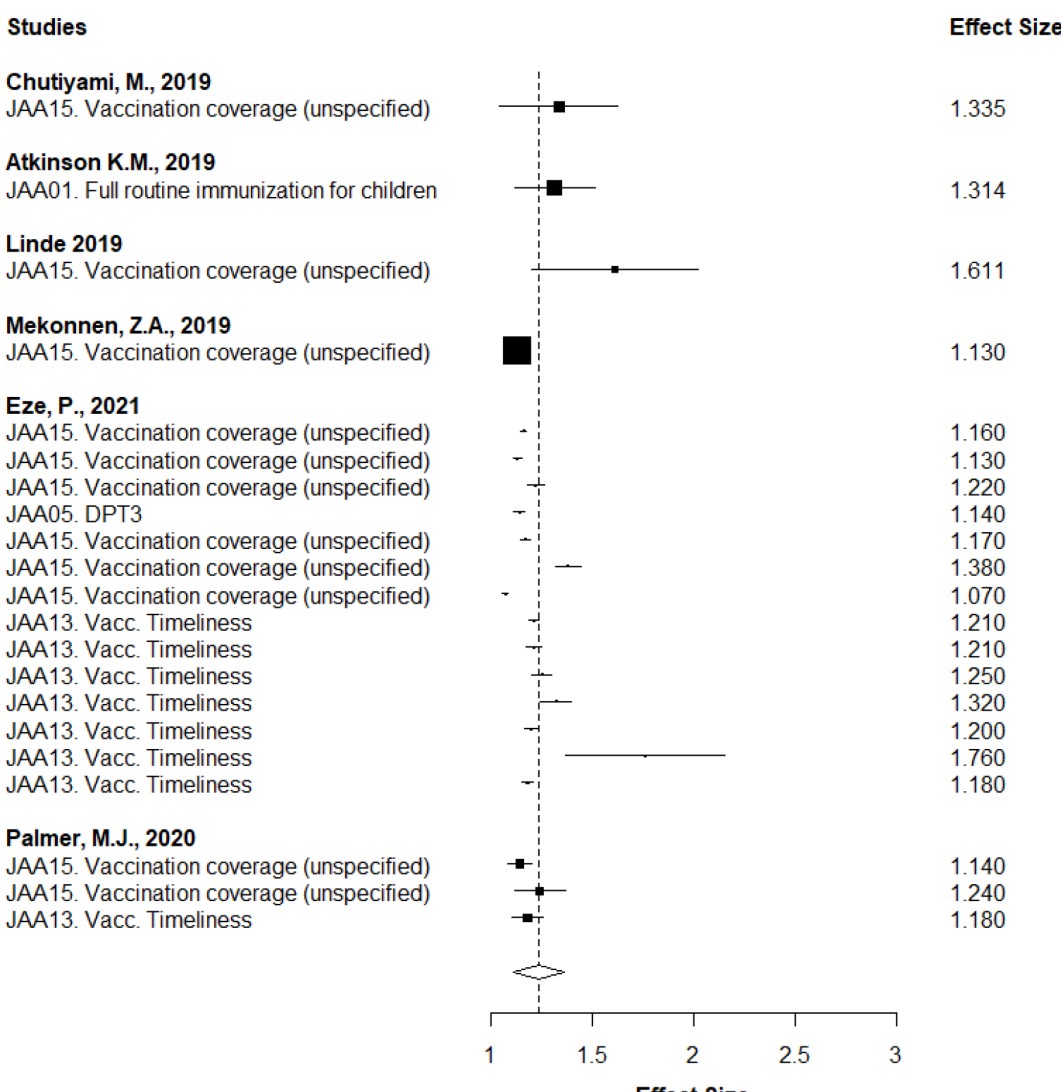

**Figure 3**   Robust variance estimation (RVE) of interventions of written or pictorial messages to caregivers (AB4), mean baseline coverage values. Intervention and outcome codes are taken from online supplemental appendix 1. DPT, diphtheria, pertussis and tetanus.

interventions in L&MICs at a higher level of abstraction (ie, at the systematic review of systematic reviews level) and this brings about a number of methodological challenges. For example, the validity of this review depends on the coverage and quality of the underlying evidence base and thus using the SURE checklist we drew attention to the mixed quality of the primary evidence base that informed the findings of our included systematic reviews. Consequently, caution must be exercised in the interpretation of the systematic review evidence, especially as no mitigating actions were taken to deal with potential biases. One of the ways to ensure robustness of evidence is to provide subgroup analysis that disaggregates by levels of confidence. However, the majority of the included systematic reviews did not provide subgroup analysis because their analyses were mostly based on small study samples that were not sufficient for credible subgroup analyses. In addition, we were faced with high levels of heterogeneity and a wide range of synthesis

approaches limiting our pool of studies for meta-analysis. We addressed these limitations to a certain extent by also considering qualitative evidence. Generally, we have dealt with these methodological challenges as much as possible following guidance provided by the Cochrane and Campbell Collaborations. Finally, most of the included studies did not pay sufficient attention to unpacking causal mechanisms which limited our ability to firmly conclude how and for whom routine child immunisation interventions are working.

### Implications for policy and practice
#### Implications for caregiver-oriented interventions
The caregiver-oriented interventions are effective. Sensitisation and education campaigns (AA2) mostly address the knowledge gaps of caregivers on the importance of vaccinations, importance of maintaining a schedule and potential misconceptions around vaccinations. This information is delivered by either frontline health workers or

trained facilitators at the health facility, at caregiver homes or in community groups. The SMS reminders (AB4) delivered to the mobile phones of caregivers address the practical barriers that they face regarding when and where to take their child for a vaccine or follow-up doses. Thus, these interventions can be effective in improving vaccine uptake in communities where such barriers are prevalent.

In the last decade, a lot of the attention of researchers, programme implementers and policymakers has been on how to motivate caregivers to vaccinate their children through monetary incentives and there have been a substantial number of experiments involving conditional and/or unconditional cash transfer schemes to assess their effectiveness in improving immunisation outcomes.[15] However, the evidence does not provide a clear answer on their effectiveness and there is a need for an updated review with a more inclusive search strategy for identifying relevant articles published in non-health journals.

### Implications for health system-oriented interventions

Among the health system-oriented interventions, home visits (BB7) are worth considering for improvement in vaccine uptake as we find them to be effective. For pay-for-performance schemes (BD5) we find mixed evidence. As a recent review[7] of this intervention category is available and of high quality, there is no need for an update and policymakers can consult it for more details.

Two intervention categories which are of specific policy relevance and would benefit from a stronger evidence base are: (a) interventions focusing on building skills, capacity and motivation of formal health workers (BA1, BA2, BD1, BD2) and (b) new Health Management Information System (HMIS)/dashboard systems (BG1), that is, interventions leveraging digital technologies, which are increasingly being adopted and expanded across L&MICs. For both these intervention categories, Engelbert et al[15] found a substantial number of primary studies, though most were not single but multicomponent interventions.

While the evidence on health system-oriented interventions in the context of immunisation is limited, policymakers seeking to strengthen health systems for the purpose of improving immunisation outcomes can consult the broader literature on health system strengthening for guidance. In fact, there is a systematic review of reviews available,[31] which synthesised evidence that assessed the effects of health systems strengthening interventions on health status and health system outcomes (service utilisation, quality service provision, uptake of healthy behaviours and financial protection) in L&MICs, which may be helpful depending on its quality.

### Implications for community-oriented interventions

For guidance on effectiveness of community-oriented interventions, the recent review by Jain et al[29 30] can be consulted as it is comprehensive, provides quantitative evidence and uses a nuanced framework to classify community-oriented interventions based on the process of engaging communities.

### Implications for research

Besides addressing critical evidence gaps as described in the section above, we also need to better understand complementarity of interventions, for example, some reviews analysed combinations of caregiver and health system-oriented intervention types (bridging the demand and supply side gaps) finding favourable but also inconclusive effects. We do not yet know which intervention combinations work best in terms of improving immunisation outcomes. Further theory development may be a starting point for contributing to a better understanding of the enablers and barriers of interventions as well as unpacking underlying causal mechanisms and thus improving the selection and targeting of immunisation programmes.

In addition, future research needs to engage more with cost-effectiveness of interventions as we need to learn about what works and at what cost to enable selection of the most impactful and most cost-effective interventions to improve routine child immunisation outcomes. Further work is also needed in improving the quality of both systematic reviews and primary studies. At the review level future research should attempt more subgroup analyses by levels of confidence in primary studies to instil more trust in the results. At the primary study level, researchers should use more robust evaluation methods to minimise the potential biases in the measurement of effects. More work is required to develop user-friendly quality assessment tools for systematic reviewers that minimise the scope for subjective judgements.

**Acknowledgements** This research has been undertaken as a part of 3ie's immunisation evidence programme, supported by the Bill and Melinda Gates Foundation, Seattle, USA. We thank Sohail Agha, previous Senior Program Officer, Gates Foundation, Seattle, for his support in conceptualisation as well as engagement on the review. Special thanks to Tove Ryman, Senior Program Officer, Gates Foundation, Seattle, whose guidance helped shape the analysis and dissemination of this review. We thank Mark Engelbert who was instrumental in guiding the search process and Avantika Bagai who provided support in the initial phase of the review. We also thank Birte Sniltsveit at 3ie for her comments on this review and Sebastian Martinez at 3ie for his overall support of the review.

**Contributors** MJ conceived the review. MJ, SS and MD wrote the protocol. SSP and DA did the systematic search. SSP and MDAL screened and identified studies and MJ and MD made the final decisions regarding study inclusion. SSP and MDAL did the data extraction and critical appraisal. SS, MDAL and MD did the statistical analysis. MD and SSP did the qualitative analysis. MD and MJ wrote the manuscript. MJ provided critical inputs on the whole analysis, checked the data, coordinated the review and had full access to all materials and results. All authors critically reviewed and revised the manuscript and approved the final document for submission. MJ is responsible for the overall content and is the guarantor of this review.

**Funding** This research was funded through a grant from the Bill & Melinda Gates Foundation (INV-008461) to the International Initiative for Impact Evaluation (3ie). Through this grant the following work has been commissioned: (a) 3ie provided funding and technical assistance for seven impact evaluations of community engagement interventions for immunisation; (b) 3ie conducted a systematic review of community engagement interventions to assess their effectiveness in improving routine child immunisation in L&MICs; and (c) 3ie conducted an evidence gap map on routine child immunisation in L&MICs.

**Disclaimer** The funder of the study had no role in the study design, data collection, data analysis, data interpretation, or writing of the report.

**Competing interests** As members of 3ie staff, MJ had been involved in reviewing deliverables for the evaluations funded by 3ie and providing research teams with technical assistance. MJ, SS, MDAL and SSP have also worked as authors on the systematic review and MJ and SSP have been coauthors of the EGM report. Among these, only the work on the systematic review could pose a potential conflict of interest. However, several procedural safeguards and transparency measures were put in place to mitigate the risk this conflict of interest imposed. All candidate systematic reviews, including the one carried out by 3ie, have undergone a rigorous screening and critical appraisal process by two independent screeners. The authors have no financial interest in this area. MD has authored systematic reviews and systematic reviews of reviews but largely on issues related to financial inclusion. MD is also an editor of the International Development Coordinating Group at the Campbell Collaboration, but this will not pose any conflicts of interest.

**Patient and public involvement** Patients and/or the public were not involved in the design, or conduct, or reporting, or dissemination plans of this research.

**Patient consent for publication** Not applicable.

**Ethics approval** The ethical approval was not required for this umbrella review as all the data were generated from published systematic reviews.

**Provenance and peer review** Not commissioned; externally peer reviewed.

**Data availability statement** Data are available upon reasonable request. The systematic reviews and other documents included in this study are public and available. The data extracted from these studies are available from the corresponding author upon request.

**ORCID iDs**
Monica Jain http://orcid.org/0000-0001-5428-377X
Shradha S Parsekar http://orcid.org/0000-0002-8824-9198

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
