## [Reviewer comments · BMJ Open]

ARTICLE DETAILS

TITLE (PROVISIONAL)	Effective interventions for improving routine childhood immunisation in low- and middle-income countries: A systematic review of systematic reviews
AUTHORS	Jain, Monica; Duvendack, Maren; Shisler, Shannon; Parsekar, Shradha; Leon, Maria Daniela

VERSION 1 – REVIEW

REVIEWER	Walden, Rachel Vanderbilt University
REVIEW RETURNED	05-Jul-2023

GENERAL COMMENTS	You will want to add the publisher/platforms that were used to search the databases to Appendix 3. example: Medline (Ovid), Embase (Elsivier), CINAHL (EbscoHost)
--

REVIEWER	Loke, Yoon University of East Anglia, Norwich Medical School
REVIEW RETURNED	21-Sep-2023

GENERAL COMMENTS	Thank you for giving me the opportunity to comment on this interesting and well written submission. My main suggestions are with regards to the style of writing, particularly in the Abstract section and the Strengths and Limitations at the beginning. I didn't feel it was appropriate that the objective was 'To provide clarity...'. The aim of such overviews is to provide a comprehensive synthesis. At the end it may or may not provide greater clarity depending on the results. It would help the reader if the abstract results paragraph could briefly state the number of included reviews, as this is crucial information for readers to know the scope of this article. Unfortunately, I felt that the strengths and limitations section at the beginning had too much subjectivity, and I feel it would be better to avoid overselling the research work. I believe it is hard to justify the words 'extensive and novel' to be claimed as strengths, because these are very subjective judgements. Both attributes may be true but there are also lots of extensive and novel works which are very poor research quality.
--

	I felt that the final sentence on limitations was non-specific, and failed to explain how the major limitations affected the validity or reliability of the results. Generally, I would like the limitations section to highlight appropriate cautions for readers when reading the results, and to explore how certain biases could have swayed the findings one way or the other. I am aware that in the main text, the limitations section is similarly weak, and I will encourage the authors to provide a more detailed identification of the major limitations. I don't believe that either of the sections should claim 'unique methodological challenges'. I found the section on implications for practise and policy to be too long to comprehend. I hope that subsections can be considered
--	--

REVIEWER	Fukuda, Yoshiharu Yamaguchi University, Community Health and Medicine
REVIEW RETURNED	18-Nov-2023

GENERAL COMMENTS	The strength of this paper is the use of intervention taxonomy. This should be mentioned in the abstract. Taxonomy is listed in Table 2, but it is difficult to read. We recommend creating a table with only Intervention Taxonomy, or only Taxonomy and the number of SR. Instead, taxonomies with no SR in Table 2 could be omitted.
---

VERSION 1 – AUTHOR RESPONSE

Reviewer: 1: Miss Rachel Walden, Vanderbilt University	You will want to add the publisher/platforms that were used to search the databases to Appendix 3. example: Medline (Ovid), Embase (Elsivier), CINAHL (EbscoHost)	Appendix 3	As suggested, we have added the platforms that were used for searching databases mentioned in Appendix 3.
Reviewer: 2, Prof. Yoon Loke, University of East Anglia	My main suggestions are with regards to the style of writing, particularly in the Abstract section and the Strengths and Limitations at the beginning. I didn't feel it was appropriate that the objective was 'To provide clarity...'. The aim of such overviews is to provide a comprehensive synthesis. At the end it may or may not provide greater clarity depending on the results. It would help the reader if the abstract results paragraph could briefly state the number of included reviews, as this is crucial	Abstract	We have taken this comment on board. We tweaked the text in the abstract section in relation to the main objective, we also added the number of included reviews to the results section in the abstract. Furthermore, we re-wrote the strengths and limitations section at the beginning and also in the main text, see our more detailed response on the latter below.

	information for readers to know the scope of this article.		
	Unfortunately, I felt that the strengths and limitations section at the beginning had too much subjectivity, and I feel it would be better to avoid overselling the research work. I believe it is hard to justify the words 'extensive and novel' to be claimed as strengths, because these are very subjective judgements. Both attributes may be true but there are also lots of extensive and novel works which are very poor research quality. I felt that the final sentence on limitations was non-specific, and failed to explain how the major limitations affected the validity or reliability of the results. Generally, I would like the limitations section to highlight appropriate cautions for readers when reading the results, and to explore how certain biases could have swayed the findings one way or the other. I am aware that in the main text, the limitations section is similarly weak, and I will encourage the authors to provide a more detailed identification of the major limitations. I don't believe that either of the sections should claim 'unique methodological challenges'.	Strengths & limitations	We thank the reviewer for his comment. We have removed the words 'extensive and novel' and entirely rewritten the strengths and limitations section at the beginning but also in the main text to provide more detail on the major limitations of this review and how these limitations may affect the validity of our work. We hope our additional text satisfies the reviewer.
	I found the section on implications for practise and policy to be too long to comprehend. I hope that subsections can be considered	Implications for practice and policy	We thank the reviewer for this comment. As suggested, we have reduced the length of this section and divided this section into sub-sections for ease of comprehension.
Reviewer 3, Dr. Yoshiharu Fukuda,	The strength of this paper is the use of intervention taxonomy. This should be mentioned in the abstract. Taxonomy is listed in Table 2, but it is difficult to read. We recommend creating a table	Abstract, Table 2	We agree that Table 2 is complex and we have thus simplified it as per the suggestions made by the reviewer, and removed the intervention categories for

Yamaguchi University	with only Intervention Taxonomy, or only Taxonomy and the number of SR. Instead, taxonomies with no SR in Table 2 could be omitted.		which we have no evidence. We have also removed the quantitative and qualitative findings columns. The original table has been shifted as supplementary file (Appendix 9)
---	--	---

VERSION 2 – REVIEW

REVIEWER	Loke, Yoon University of East Anglia, Norwich Medical School
REVIEW RETURNED	25-Jan-2024

GENERAL COMMENTS	Thank you for the much-improved manuscript.
---